# The CHA_2_DS_2_-VASc Score Predicts New-Onset Atrial Fibrillation and Hemodynamic Complications in Patients with ST-Segment Elevation Myocardial Infarction Treated by Primary Percutaneous Coronary Intervention

**DOI:** 10.3390/diagnostics12102396

**Published:** 2022-10-01

**Authors:** Dan Alexandru Cozac, Eva Katalin Lakatos, Zoltan Demjen, Alexandru Ceamburu, Paul Ciprian Fișcă, Ioana Șuș, Laszlo Hadadi, Alina Scridon

**Affiliations:** 1Department of Physiology, University of Medicine, Pharmacy, Science and Technology “George Emil Palade” of Târgu Mureș, 540142 Târgu Mureș, Romania; 2Emergency Institute for Cardiovascular Diseases and Transplantation Târgu Mureș, 540142 Târgu Mureș, Romania

**Keywords:** atrial fibrillation, CHA_2_DS_2_-VASc score, hemodynamic complications, prediction, ST-segment elevation myocardial infarction

## Abstract

Arrhythmic and hemodynamic complications related to ST-segment elevation myocardial infarction (STEMI) represent a major clinical challenge. Several scores have been developed to predict mortality in STEMI. However, those scores almost exclusively include factors related to the acute phase of STEMI, and no score has been evaluated to date for its ability to specifically predict arrhythmic and hemodynamic complications. We, thus, aimed to assess the ability of chronic risk factors burden, as expressed by the CHA_2_DS_2_-VASc score, to predict STEMI-related arrhythmic and hemodynamic complications. Data were collected from 839 consecutive STEMI patients treated by primary percutaneous coronary interventions (pPCI). CHA_2_DS_2_-VASc and GRACE scores were calculated for all patients, and their ability to predict STEMI-related arrhythmic (i.e., new-onset atrial fibrillation (AF), ventricular tachycardia/fibrillation) and hemodynamic (i.e., cardiogenic shock, asystole) complications was assessed in univariate and multiple regression analysis. Arrhythmic and hemodynamic complications occurred in 14.8% and 10.2% of patients, respectively. Although the GRACE score outweighed the CHA_2_DS_2_-VASc score in the ability to predict STEMI-related hemodynamic complications (*p* < 0.0001), both scores had a similar predictive value for STEMI-related new-onset AF (*p* = 0.20), and both remained independent predictors of new-onset AF and of hemodynamic complications in the multiple regression analyses. A CHA_2_DS_2_-VASc score > 2 points independently predicted new-onset AF (*p* < 0.01) and hemodynamic complications (*p* = 0.04). Alongside the GRACE score, the CHA_2_DS_2_-VASc score independently predicted new-onset AF and hemodynamic complications in STEMI patients treated by pPCI. These data suggest that a combination of acute and chronic risk factors could provide additional benefit in identifying patients at risk of STEMI-related complications, who could benefit from closer follow-up and more intensive prophylactic and therapeutic strategies.

## 1. Introduction

ST-segment elevation myocardial infarction (STEMI) is one of the leading causes of mortality and morbidity worldwide [1]. Despite major advancements in treatment strategies over the last decades, arrhythmic and hemodynamic complications in the acute phase of STEMI remain a major cause of early mortality [2,3]. Early identification of patients who will develop post-STEMI hemodynamic and arrhythmic complications could allow earlier and more intensive implementation of monitoring, prophylactic, and therapeutic strategies and could, thus, improve cardiovascular prognosis in patients with STEMI. A series of markers and risk scores have been proposed to allow risk stratification in patients with STEMI [4,5,6,7]. However, these scores almost exclusively include risk factors characteristic to the acute phase of STEMI. Other, preexisting factors have also been shown, however, to impair prognosis in STEMI patients [8,9,10,11].

The CHA_2_DS_2_-VASc score is the most widely used clinical score for estimating the risk of stroke in patients with atrial fibrillation (AF) [12]. However, the CHA_2_DS_2_-VASc score can also be seen as a surrogate for cardiovascular risk burden in the general population. Studies in patients with STEMI undergoing primary percutaneous coronary interventions (pPCI) have recently linked this score with various STEMI-related complications, including the “no-reflow” phenomenon and contrast-induced nephropathy [13,14,15]. A higher CHA_2_DS_2_-VASc score has also been associated with a higher risk of in-hospital and long-term mortality risk in patients with STEMI [16]. However, since arrhythmic and hemodynamic complications benefit from different monitoring, prevention, and treatment methods, predicting these complications separately would be of great clinical interest. To date, no study has evaluated the ability of the CHA_2_DS_2_-VASc score to specifically predict early post-STEMI arrhythmic and hemodynamic complications.

Accordingly, the present study aimed to evaluate the ability of the CHA_2_DS_2_-VASc score to predict the occurrence of STEMI-related arrhythmic and hemodynamic complications in patients treated by pPCI.

## 2. Materials and Methods

### 2.1. Study Population

Data were retrospectively collected for consecutive patients treated by pPCI for STEMI at the Emergency Institute for Cardiovascular Diseases and Transplantation of Târgu Mureș, Romania, between January 2011 and December 2017. The research protocol complied with the Declaration of Helsinki and was approved by the local Ethics Committee. All patients, with diagnostic criteria for STEMI according to the European Society of Cardiology Guidelines for the Management of Acute Myocardial Infarction in Patients Presenting with ST-segment Elevation [17], who were treated by pPCI within the first 12 h after symptoms onset or 12–24 h after symptoms onset if they presented evidence of ongoing myocardial ischemia, were included in the study. Patients with pre-existing AF were excluded from the study.

### 2.2. Evaluated Parameters

The components of the CHA_2_DS_2_-VASc score (i.e., congestive heart failure, hypertension, age ≥ 75 years, diabetes mellitus, stroke, vascular disease, age 65–74 years, sex) were recorded, and a CHA_2_DS_2_-VASc score was calculated for each patient included in the study. Several other cardiovascular risk factors identified as predictors of complications in STEMI patients in previous studies [10,11,18] that are not included in the CHA_2_DS_2_-VASc score were also evaluated, including smoking status, heart rate and left ventricular ejection fraction (LVEF) on admission, anterior location of myocardial infarction, history of previous myocardial infarction, chronic kidney disease, and chronic respiratory disease.

### 2.3. Outcome Definitions

The occurrence of arrhythmic and hemodynamic complications was assessed in all patients during their hospital stay. All patients underwent continuous ECG monitoring during the first 96 h following the pPCI procedure. Intermittent regular and symptom-driven ECG recordings were performed for the rest of the hospital stay.

Arrhythmic complications evaluated in the present study included new-onset AF, ventricular tachycardia, and ventricular fibrillation. New-onset AF was defined as at least one episode of AF (>30 s) occurring during post-STEMI hospital stay [12]. Ventricular tachycardia was defined as wide QRS complex (>120 ms) tachycardia, with key electrocardiographic criteria supporting the ventricular origin of the tachyarrhythmia [19]. Ventricular fibrillation was defined as disorganized cardiac electrical activity resulting in cardiac arrest and loss of consciousness.

Hemodynamic complications included cardiogenic shock and asystole. Even if the latter represents a cardiac electrical disorder, if it occurred with predilection in patients with severe impairment of cardiac mechanical activity, it was, therefore, included in the present study as a marker of hemodynamic impairment. Cardiogenic shock was defined as persistent arterial hypotension (systolic blood pressure < 90 mmHg) refractory to volume resuscitation, with features of end-organ hypoperfusion, or need for inotropic/vasoactive medication to maintain adequate perfusion pressure [17]. Asystole was defined as lack of any electrical activity on the surface ECG.

### 2.4. Statistical Analyses

All analyzed parameters presented non-normal distribution. Continuous variables are, thus, expressed as median (interquartile range). Categorical variables are expressed as absolute values and percentages. Fischer’s exact test was used to compare categorical data, and the Mann–Whitney U test was used to analyze continuous variables. All factors that were significantly different (*p* < 0.05) between groups and factors identified as predictors of STEMI-related complications in previous studies were included in the univariate logistic regression analysis. Statistically significant variables (*p* < 0.05) identified using the univariate regression analysis were then included in the multiple regression analysis, to identify independent predictors of the studied complications. Receiver operating characteristic (ROC) analysis and the area under the curve (AUC) were used to identify the cut-off values of the CHA_2_DS_2_-VASc score that best predicted the study outcomes. Comparison of ROC curves was used to identify the variables with the highest discriminatory power. All tests were two-sided, and a *p*-value < 0.05 was considered statistically significant. All data were computed using JASP 0.14.1 (GNU Affero General Public License, The Netherland) and MedCalc 19.6.4 (MedCalc Software, Belgium).

## 3. Results

A total of 839 patients treated by pPCI for STEMI were included in the analysis. The median duration of follow-up was 8 (7–10) days. Of the 839 patients, 126 (14.8%) patients presented arrhythmic complications (2.4% presented new-onset AF, and 13.3% presented ventricular tachycardia/fibrillation). Hemodynamic complications were recorded in 87 (10.2%) of the study patients (9.0% presented cardiogenic shock, and 4.7% presented asystole). In-hospital death occurred in 50 (5.95%) of the study patients.

### 3.1. Predictors of ST-Segment Elevation Myocardial Infarction-Related Arrhythmic Complications

Compared with their non-arrhythmic counterparts, patients that presented STEMI-related arrhythmic complications (Table 1) were older (*p* = 0.01), had lower blood pressure (*p* < 0.001) and lower LVEF (*p* < 0.001) on admission, and were more likely to present CKD (*p* < 0.01).

Gender distribution, anterior location and history of previous myocardial infarction, and presence of diabetes mellitus, arterial hypertension, chronic heart failure, and chronic respiratory disease were all similar between patients with and without post-STEMI arrhythmic complications (all *p* >0.05). Similarly, there was no significant difference in the CHA_2_DS_2_-VASc scores between the two groups (*p* = 0.37).

In univariate regression analysis, age, systolic blood pressure and LVEF on admission, and the presence of CKD were significantly associated with an increased risk of post-STEMI arrhythmic complications (all *p* < 0.01). There was no significant association between the CHA_2_DS_2_-VASc score and the occurrence of arrhythmic complications in univariate regression analysis (*p* = 0.09).

In multiple logistic regression analysis, including all factors that differed significantly between patients with and without STEMI-related arrhythmic complications, age > 51 years, LVEF < 43% and systolic blood pressure < 110 mmHg on admission, and the presence of CKD remained independent predictors of STEMI-related arrhythmic complications (all *p* < 0.05) (Table 2). In ROC analysis (Figure 1A), systolic blood pressure < 110 mmHg (ΔAUC 0.62, 95% CI 0.58–0.65, *p* < 0.01) was a better predictor of STEMI-related arrhythmic complications than age > 51 years or the presence of CKD (both *p* < 0.05) and had a similar predictive value as a LVEF < 43% on admission (*p* = 0.09).

When arrhythmic complications were assessed separately, there was no significant association between the CHA_2_DS_2_-VASc score and the occurrence of ventricular arrhythmias (*p* = 0.86). Meanwhile, the CHA_2_DS_2_-VASc scores were significantly higher in patients who presented new-onset AF during their post-STEMI hospital stay than in their non-arrhythmic peers (3 (1–3) points vs. 1 (1–3) point, *p* = 0.001). In multiple logistic regression analysis, a CHA_2_DS_2_-VASc score > 2 points remained an independent predictor of post-STEMI AF (OR 3.9, 95% CI 0.4–2.2, *p* < 0.01), alongside the presence of CKD (OR 5.4, 95% CI 0.7–2.6, *p* < 0.001). Moreover, when compared with a GRACE score >196 points, a CHA_2_DS_2_-VASc score > 2 points had a similar ability to predict post-STEMI new-onset AF (ΔAUC 0.06, 95% CI 0.03–0.10, *p* = 0.20). In addition, when both scores were included in the multiple logistic regression analysis, both remained independent predictors of post-STEMI new-onset AF (OR 3.4, 95% CI 1.3–8.7, *p* < 0.01 and OR 3.1, 95% CI 1.1–8.8, *p* = 0.02, respectively).

### 3.2. Predictors of ST-SEGMENT Elevation Myocardial Infarction-Related Hemodynamic Complications

Patients who presented hemodynamic complications during hospitalization for STEMI (Table 3) were significantly older and more frequently presented CKD (both *p* < 0.0001).

Gender distribution, anterior location and history of previous myocardial infarction, and presence of diabetes mellitus, arterial hypertension, chronic heart failure, and chronic respiratory disease were all similar between patients with and without STEMI-related hemodynamic complications (all *p* > 0.05). The CHA_2_DS_2_-VASc scores were also similar between the two groups (*p* = 0.51).

However, in univariate regression analysis, age, history of CKD, and the CHA_2_DS_2_-VASc score had a significant predictive value for the occurrence of cardiogenic shock or asystole (all *p* < 0.05). Moreover, the presence of CKD and a CHA_2_DS_2_-VASc score > 2 points remained strong independent predictors of STEMI-related hemodynamic complications in multiple logistic regression analysis (both *p* < 0.05) (Table 4). Both factors displayed a similar ability to predict cardiogenic shock or asystole among patients with STEMI (ΔAUC 0.004, 95% CI 0.08–0.09, *p* = 0.90). Compared with a CHA_2_DS_2_-VASc score > 2 points, a GRACE score > 196 points had a better ability to discriminate hemodynamic complications among patients with STEMI (*p* < 0.0001) (Figure 1B). However, both scores remained independent predictors of post-STEMI hemodynamic complications in multiple regression analysis (OR 1.5, 95% CI 1.0–2.5, *p* = 0.04 and OR 2.7, 95% CI 1.4–5.1, *p* = 0.001, respectively).

Comparable results were also obtained when hemodynamic complications were assessed separately. The CHA_2_DS_2_-VASc scores were significantly higher in patients who presented cardiogenic shock or asystole during their post-STEMI hospital stay than in their non-affected counterparts (2 (1–3) points vs. 1 (1–3) point, *p* = 0.001, for cardiogenic shock, and 3 (1–3) points vs. 2 (1–3) points, *p* < 0.001, for asystole). In multiple logistic regression analysis, a CHA_2_DS_2_-VASc score > 2 points remained an independent predictor of cardiogenic shock (OR 1.6, 95% CI 1.0–2.7, *p* = 0.03), as well as of asystole (OR 2.6, 95% CI 1.3–5.0, *p* < 0.001), alongside CKD (OR 3.0, 95% CI 1.6–5.5, *p* < 0.001 and OR 2.2, 95% CI 0.9–5.1, *p* = 0.04, respectively). In line with previous studies [16], a CHA_2_DS_2_-VASc score > 2 points was also associated with increased in-hospital mortality (OR 3.63, 95% CI 2.02–6.53, *p* < 0.001).

## 4. Discussion

The main findings of the present study were that (1) arrhythmic and hemodynamic complications were rather common, affecting ≈15% and ≈10% of STEMI patients treated by pPCI, respectively; and (2) CHA_2_DS_2_-VASc scores independently predicted new-onset AF and hemodynamic complications in the study patients. Although, (3) the GRACE score outweighed the CHA_2_DS_2_-VASc score in its ability to predict post-STEMI hemodynamic complications, (4) both scores had similar predictive value for STEMI-related new-onset AF, and (5) both remained independent predictors of new-onset AF and of hemodynamic complications in the multiple regression analyses.

### 4.1. Arrhythmic and Hemodynamic Complications Remain Common in the Era of Primary Percutaneous Coronary Interventions

The management of STEMI patients has seen tremendous improvements over the past decades, which is mainly related to the widespread use of mechanical and pharmacological myocardial revascularization strategies and to the advent of modern antithrombotic therapies. However, STEMI-related arrhythmic and hemodynamic complications continue to be an important source of cardiovascular morbidity and mortality in these patients. Atrial fibrillation is one of the most common arrhythmias among STEMI patients, with an incidence of new-onset AF of 2.4% in the present study. This incidence is lower than that reported in previous studies, such as the Assessment of Pexelizumab in Acute Myocardial Infarction (APEX-AMI) or VALsartan in Acute Myocardial Infarction (VALIANT) trials, where the incidence of AF was reported as between 6% and 12% [20,21]. This difference can be explained, however, by the fact that many of the patients included in the above-mentioned trials had preexisting AF [20,21], and new-onset AF was not specifically evaluated in those trials. The incidence of ventricular tachycardia/fibrillation was, on the other hand, 13.3% in the present study, considerably higher than that reported in the large APEX-AMI and the Air Primary Angioplasty in Myocardial Infarction (PAMI) trials, where the incidence was 5.7% [20,22]. Patients with isolated inferior STEMI, those presenting more than six hours after symptoms onset, and those with CKD were, however, excluded from those trials, which could explain the differences recorded in ventricular arrhythmias incidence. Meanwhile, the incidences of cardiogenic shock (9.0%), and that of asystole (4.7%) recorded in the present cohort were in line with those reported in several previous trials [17,23,24].

Data from the SWEDEHEART registry showed that the morbidity and mortality of patients with acute coronary syndromes have decreased considerably over the last decades. However, this decline does not seem to apply to the population with STEMI-related complications [25]. Therefore, identifying strategies that would allow for the prediction of major STEMI-related complications as early as possible remains a critical issue, including in the era of pPCI. Since monitoring, prophylactic, and therapeutic methods can be considerably different for arrhythmic and hemodynamic complications, the development of strategies that allow separate risk stratification for these STEMI-related complications would be extremely important for appropriate patient management.

### 4.2. CHA_2_DS_2_-VASc Score above 2 Points Independently Predicts New-Onset Atrial Fibrillation in Patients with ST-Segment Elevation Myocardial Infarction

In the present study, systolic blood pressure < 110 mmHg and LVEF < 43% on admission, age > 51 years, and presence of CKD independently predicted the occurrence of STEMI-related arrhythmic complications. Meanwhile, in line with the recent data reported by Crenshaw et al. [26], other classic cardiovascular risk factors known to promote proarrhythmic atrial electrical and/or structural remodeling, such as hypertension, heart failure, diabetes mellitus, or chronic respiratory disease [27,28,29], many of which are included in the CHA_2_DS_2_-VASc score, had no significant impact on the occurrence of post-STEMI arrhythmias. Consequently, the CHA_2_DS_2_-VASc score had no predictive value for post-STEMI arrhythmias. Meanwhile, the GRACE score independently predicted both ventricular and atrial post-STEMI arrhythmias. The inability of the CHA_2_DS_2_-VASc score to predict post-STEMI arrhythmic complications was related, however, to its lack of association with post-STEMI ventricular arrhythmias. Meanwhile, a CHA_2_DS_2_-VASc score > 2 points was a strong independent predictor of post-STEMI new-onset AF. Moreover, in the multiple regression analysis, both the CHA_2_DS_2_-VASc and the GRACE scores remained independent predictors of post-STEMI new-onset AF. Together, these data indicate that the factors related to the acute phase of STEMI (included in the GRACE score) play central roles in the occurrence of ventricular arrhythmias, whereas in this setting the role of chronic risk factors (included in the CHA_2_DS_2_-VASc score) is likely to be negligible. Meanwhile, both chronic and acute atrial changes seem to contribute to the occurrence of STEMI-related atrial arrhythmias. It is now widely accepted that the pathophysiology of AF relies on a combination of ectopic triggers that initiate the arrhythmia and an arrhythmogenic substrate that ensures the formation of reentry circuits and, hence, the persistence of the arrhythmia [27]. In STEMI, factors such as acute ischemia, myocardial inflammation, sympathetic overactivation, metabolic and electrolyte imbalance, and hemodynamic impairment have been related to a significant increase in atrial ectopic activity, whereas the burden of chronic cardiovascular risk factors, such as those included in the CHA_2_DS_2_-VASc score, are known to promote atrial proarrhythmic electrical and structural remodeling [27,28,30]. The combination of chronic and acute risk factors and its ability to provide the optimal environment for AF to occur could, thus, explain the ability of both the CHA_2_DS_2_-VASc and GRACE scores to independently predict STEMI-related new-onset AF.

### 4.3. CHA_2_DS_2_-VASc Score above 2 Points Independently Predicts Hemodynamic Complications in Patients with ST-Segment Elevation Myocardial Infarction

In the present study, a CHA_2_DS_2_-VASc score > 2 points and the presence of CKD were independent predictors of STEMI-related hemodynamic complications (i.e., cardiogenic shock or asystole). Meanwhile, factors included in the CHA_2_DS_2_-VASc score, such as arterial hypertension, diabetes mellitus, and chronic heart failure, were not associated with a higher risk of post-STEMI hemodynamic complications in the present cohort. Together, these results reinforce the importance of a combination of cardiovascular factors that, together, exert more accentuated myocardial damage than the presence of each factor alone.

In ROC analysis, CKD and a CHA_2_DS_2_-VASc score > 2 points had similar predictive values for the occurrence of STEMI-related hemodynamic complications, emphasizing the major cardiovascular impact of CKD, which appears to be comparable to that of a combination of other classic cardiovascular risk factors, as illustrated by the CHA_2_DS_2_-VASc score. In the present study, a GRACE score > 196 points was a better predictor of cardiogenic shock or asystole than a CHA_2_DS_2_-VASc score > 2 points, suggesting that although chronic, pre-existing conditions may be highly relevant for the occurrence of STEMI-related hemodynamic complications, in this setting, the acute changes reflected in the GRACE score could play a more important role. A CHA_2_DS_2_-VASc score > 2 points remained, however, an independent predictor of hemodynamic complications, even after correction for the GRACE score. These results strongly support the potential usefulness of a risk score that combines elements related to chronic disease burden (reflected by elements of the CHA_2_DS_2_-VASc score) and factors related to the acute phase of STEMI (expressed by components of the GRACE score).

### 4.4. Clinical Implications

The GRACE score is a well-known, prospectively validated scoring system that estimates in-hospital and 6-month mortality rates in patients with acute coronary syndromes. Our results add to the already existing data, showing that the GRACE score is also capable of independently and specifically predicting STEMI-related arrhythmic and hemodynamic complications. Moreover, the present study also identified the CHA_2_DS_2_-VASc score as an independent predictor of new-onset AF and hemodynamic complications, even after correction for the GRACE score. These findings strongly endorse the potential benefit of a combined risk score, which includes features related to chronic cardiovascular disease burden and factors associated with the acute phase of STEMI, to accurately predict these complications in patients with STEMI. If applied in routine clinical practice, such a tool would allow for the identification of patients who would benefit from closer follow-up and more intensive prophylactic and treatment strategies, which is expected to translate into a significant reduction in STEMI-related mortality.

### 4.5. Strengths and Limitations

The present study included a relatively large number of patients for which a comprehensive evaluation of numerous chronic and acute risk factors was performed. Moreover, the results obtained for the CHA_2_DS_2_-VASc score in the present study were compared with those obtained for the most widely used and validated risk score for patients with acute coronary syndromes—the GRACE score. Our results, therefore, expand the results of the previous studies in this area, showing the potential benefit of the combined usage of both the CHA_2_DS_2_-VASc and the GRACE scores to accurately predict the occurrence of new-onset AF and of hemodynamic complications in patients with STEMI. Several potential limitations of this study are also worth considering. First, our study has all the limitations of an observational study, and, therefore, cannot ascribe causality. Second, due to the retrospective nature of the study, AF presence prior to hospitalization for STEMI was only evaluated based on patient records and previous ECG tracings. The presence of AF prior to hospitalization for STEMI could not be thoroughly assessed using accurate methods such as long-term ECG monitoring. Thus, we cannot exclude the presence of asymptomatic AF prior to STEMI occurrence in some of the study patients. Although all patients underwent continuous ECG monitoring during the first 96 h following the pPCI procedure, only intermittent ECG monitoring was performed for the rest of the hospital stay. Thus, short, asymptomatic arrhythmia episodes could have escaped detection. Data regarding the type of myocardial revascularization (i.e., full or partial), which may have had an impact on the occurrence of post-STEMI complications, were also not available for the study patients.

## 5. Conclusions

The present study showed that STEMI-related hemodynamic and arrhythmic complications remain rather common in the era of pPCI. In the present cohort, the CHA_2_DS_2_-VASc score, a marker of cardiovascular risk burden, independently predicted the occurrence of new-onset AF and hemodynamic complications (cardiogenic shock or asystole). These data strongly support the need for novel risk scores, which combine chronic and acute STEMI-related risk factors, to accurately identify patients at risk of such complications, who could benefit from closer follow-up and more intensive, targeted prophylactic and therapeutic strategies.

## Figures and Tables

**Figure 1 diagnostics-12-02396-f001:**
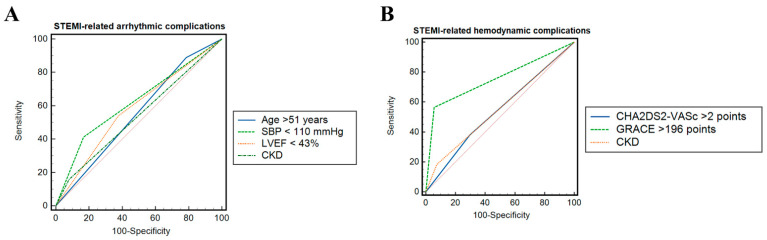
Comparison of receiver operating characteristic curves for predictors of ST-segment elevation myocardial infarction-related (**A**) arrhythmic complications and (**B**) hemodynamic complications. CKD—chronic kidney disease; LVEF—left ventricular ejection fraction; SBP—systolic blood pressure.

**Table 1 diagnostics-12-02396-t001:** Characteristics of patients with and without ST-segment elevation myocardial-infarction-related arrhythmic complications.

Parameter	Total (*n* = 831)	Arrhythmic Complications (*n* = 126)	No Arrhythmic Complications (*n* = 705)	*p*-Value	OR	95% CI
Age (years)	62 (53–70)	64 (56–74)	61 (53–70)	**0.01**	-	-
Female gender (*n*, %)	246 (29.6%)	36 (28.5%)	210 (29.7%)	0.83	0.94	0.62–1.43
**Cardiovascular risk factors**						
Active smoker (*n*, %)	390 (46.9%)	59 (46.8%)	331 (46.9%)	0.99	0.99	0.68–1.45
Arterial hypertension (*n*, %)	530 (63.7%)	81 (64.2%)	449 (63.6%)	0.92	1.02	0.69–1.52
Diabetes mellitus (*n*, %)	191 (22.9%)	24 (19.0%)	167 (23.6%)	0.30	0.75	0.47–1.22
Chronic kidney disease (*n*, %)	76 (9.1%)	20 (15.8%)	56 (7.9%)	**<0.01**	2.18	1.26–3.79
Chronic heart failure (*n*, %)	109 (13.1%)	21 (16.6%)	88 (12.4%)	0.19	1.40	0.83–2.35
Chronic respiratory diseases (*n*, %)	68 (8.1%)	15 (11.9%)	53 (7.5%)	0.11	1.66	0.90–3.05
Previous myocardial infarction (*n*, %)	63 (7.5%)	9 (7.1%)	54 (7.6%)	0.99	0.92	0.44–1.93
CHA_2_DS_S_-VASc score (points)	2 (1–3)	2 (1–3)	2 (1–3)	0.37	-	-
**Factors related to the acute phase of STEMI**						
Heart rate at admission (bpm)	80 (67–90)	78 (60–91)	80 (67–90)	0.74	-	-
Systolic blood pressure (mmHg)	130 (114–146)	122 (103–140)	130 (115–149)	**<0.001**	-	-
Left ventricular ejection fraction (%)	45 (40–45)	42 (39–45)	45 (40–47)	**<0.001**	-	-
Anterior myocardial infarction (*n*, %)	346 (41.6%)	46 (36.5%)	300 (42.5%)	0.23	0.77	0.52–1.14
**Characteristics of coronary artery disease**						
Multivessel disease (*n*, %)	524 (63.0%)	75 (59.5%)	449 (63.6%)	0.36	0.83	0.57–1.23
Culprit vessel (*n*, %)						
Left anterior descending artery	385 (46.3%)	64 (50.7%)	321 (45.5%)	0.39	-	-
Right coronary artery	334 (40.1%)	56 (44.4%)	278 (39.4%)			
Circumflex artery	112 (13.4%)	6 (4.7%)	106 (15.0%)			

Quantitative data are expressed as median (interquartile range). Categorical data are expressed as numbers (percentages). *p*-values were obtained using the Mann–Whitney U test for continuous variables and Fisher’s exact test for categorical data. Bold values indicate the parameters for which differences between groups were statistically significant (*p* < 0.05). Due to missing data regarding arrhythmic complications, 8 of the 839 study patients were excluded from all arrhythmic complications analyses. bpm—beats per minute; STEMI—ST-segment elevation myocardial infarction.

**Table 2 diagnostics-12-02396-t002:** Multiple logistic regression analysis to predict the likelihood of ST-segment elevation myocardial-infarction-related arrhythmic complications.

Variable	OR	95% CI	*p*-Value
**Model 1**			
Age > 51 years	2.35	1.27–4.34	**<0.001**
Systolic blood pressure < 110 mmHg on admission	3.75	2.44–5.75	**<0.001**
Left ventricular ejection fraction < 43% on admission	2.15	1.40–3.31	**<0.001**
Chronic kidney disease	1.93	1.07–3.50	**0.02**
Anterior myocardial infarction	0.66	0.42–1.04	0.07
**Model 2**			
CHA_2_DS_2-_VASc score > 2 points	1.17	0.46–1.35	0.40
Chronic kidney disease	2.18	1.25–3.90	**<0.001**
Anterior myocardial infarction	0.76	0.51–1.14	0.17

Cut-off values for age, systolic blood pressure and left ventricular ejection fraction on admission, and CHA_2_DS_2_-VASc score were established using receiver operating characteristic analysis. Bold values indicate the parameters for which *p*-values were statistically significant (*p* < 0.05).

**Table 3 diagnostics-12-02396-t003:** Characteristics of patients with and without ST-segment elevation myocardial-infarction-related hemodynamic complications.

Parameter	Total (*n* = 839)	Hemodynamic Complications (*n* = 87)	No Hemodynamic Complications (*n* = 752)	*p*-Value	OR	95% CI
Age (years)	62 (54–70)	66 (56–74)	61 (53–70)	**<0.0001**	-	-
Female gender (*n*, %)	236 (28.1%)	26 (29.8%)	210 (27.9%)	0.70	1.10	0.67–1.78
**Cardiovascular risk factors**						
Active smoker (*n*, %)	392 (46.7%)	35 (40.2%)	357 (47.4%)	0.21	0.74	0.47–1.17
Arterial hypertension (*n*, %)	542 (64.6%)	61 (70.1%)	481 (63.9%)	0.28	1.32	0.81–2.14
Diabetes mellitus (*n*, %)	189 (22.5%)	24 (27.5%)	165 (21.9%)	0.22	1.35	0.82–2.23
Chronic kidney disease (*n*, %)	76 (9.1%)	18 (20.6%)	58 (7.7%)	**<0.0001**	3.12	1.74–5.59
Chronic heart failure (*n*, %)	100 (11.9%)	16 (18.3%)	84 (11.1%)	0.05	1.79	0.99–3.22
Chronic respiratory diseases (*n*, %)	67 (8.1%)	11 (12.6%)	56 (7.4%)	0.09	1.79	0.90–3.58
Previous myocardial infarction (*n*, %)	61 (7.2%)	6 (6.8%)	55 (7.3%)	0.99	0.93	0.39–2.24
CHA_2_DS_S_-VASc score (points)	2 (1–3)	2 (1–3)	2 (1–3)	0.51	-	-
**Factors related to the acute phase of STEMI**						
Anterior myocardial infarction (*n*, %)	352 (41.9%)	38 (43.6%)	314 (41.7%)	0.73	1.08	0.69–1.69
**Characteristics of coronary artery disease**						
Multivessel disease (*n*, %)	528 (62.9%)	60 (68.9%)	468 (62.2%)	0.24	1.34	0.83–2.20
Culprit vessel (*n*, %)						
Anterior descending artery	404 (48.1%)	67 (77.0%)	337 (44.8%)	0.30	-	-
Right coronary artery	322 (38.3%)	17 (19.5%)	305 (40.5%)			
Circumflex artery	113 (13.4%)	3 (3.4%)	110 (14.6%)			

Quantitative data are expressed as median (interquartile range). Categorical data are expressed as numbers (percentages). *p*-values were obtained using the Mann–Whitney U test for continuous variables and Fisher’s exact test for categorical data. Bold values indicate the parameters for which differences between groups were statistically significant (*p* < 0.05). bpm—beats per minute; STEMI—ST-segment elevation myocardial infarction.

**Table 4 diagnostics-12-02396-t004:** Multiple logistic regression analysis to predict the likelihood of ST-segment elevation myocardial-infarction-related hemodynamic complications.

Variable	OR	95%	*p*-Value
CHA_2_DS_2_-VASc score > 2 points	1.59	1.00–2.52	**0.04**
Chronic kidney disease	2.95	1.64–5.32	**<0.001**

Bold values indicate the parameters for which differences between groups were statistically significant (*p* < 0.05).

## Data Availability

The datasets used and analyzed during the current study are available from the corresponding author on reasonable request.

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
