# Peer review of "The CHA2DS2-VASc Score Predicts New-Onset Atrial Fibrillation and Hemodynamic Complications in Patients with ST-Segment Elevation Myocardial Infarction Treated by Primary Percutaneous Coronary Intervention"

_diagnostics, 2022, doi:10.3390/diagnostics12102396_

Round 1

Reviewer 1 Report

In the presented manuscript, Cozac  and co-workers aimed to evaluate the ability of CHA2DS2-VASc score to predict  arrhythmic and hemodynamic complications in patients with STEMI undergoing primary PCI. It is obvious that the authors put a great deal of effort into the manuscript.  However, there are several caveats that should be corrected:

1.     Introduction is too long and should be intensively trimmed.

2.     Methods:

a)     It should be clearly stated that it is a retrospective analysis, despite the fact that  this is finally mentioned in limitations.

b)     Timeframe for  the occurrence of arrhythmic and hemodynamic complications is not provided. It can be only speculated that individuals were assessed as inpatients  as  “New-onset AF was defined as at least one episode of AF (>30 s) occurring during post-STEMI hospital stay”. This information should be provided in details along with the mean number of observation days per patient.

c)     How AF, VT and VF were detected? Did the authors apply a continuous ECG monitoring throughout hospital stay or rather intermittent regular ECG or holter ECG monitoring was used?  It is obvious that the longer ECG monitoring the larger number of arrhythmia burden can be detected.  It could be a major limitation of the study.

d)     No exclusion criteria are provided. However it is clear that if assessing new onset AF, all patient with the previous history of AF should be excluded. Moreover patients with previous MI or HF with reduced left ventricular ejection fraction are likely to develop ventricular arrhythmias anytime, not necessarily at the time of new ischaemic episode. Therefore if assessing new onset ventricular arrhythmias, all patients with previous MI or HF with reduced left ventricular ejection fraction should be excluded. According to the table 1, 7.5% of patients presented previous MI. It is  another limitation of the study.

e)     The detailed information about coronary artery disease is missing such as: the number of patients with a single and mutivessel disease, the distribution of infarct related artery and affected myocardial areas (not limited to the anterior MI),   the number of patients who underwent full and partial revascularization. Especially, the latter issue could have had a great impact on the occurrence of arrhythmic and hemodynamic complications following STEMI. It could be another major limitation of the study.

3.     Results

f)       CHA2DS2-VASc score >2 points was an independent predictor of post-STEMI AF, cardiogenic shock and asystole. Did it also predict cardiovascular mortality? What was STEMI-related mortality rate in the study cohort during monitoring period?

g)     The aim of the study was to evaluate patients presenting with STEMI. Therefore  I do not understand why Table 1 and 3 provide characteristics of patients with and without ST-segment elevation myocardial infarction.

h)     The study cohort consisted of 839 patients in total. However in the table 1 the total number of patients is  831 (126 with arrhythmic events and 705 without). Please explain this.

4.     Conclusions are too long and repeat the same sentences presented  in the results and discussion. It should be more concise and rewritten.   

Reviewer 2 Report

The CHA2DS2-VASc score predicts new-onset atrial fibrillation and hemodynamic complications in patients with ST-segment elevation myocardial infarction treated by primary percutane-ous coronary intervention

Manuscript entitled “The CHA2DS2-VASc score predicts new-onset atrial fibrillation and hemodynamic complications in patients with ST-segment elevation myocardial infarction treated by primary percutane-ous coronary intervention” by Dan et al., is a good study. Here, the authors evaluate the ability of the CHA2DS2-VASc 79 score to predict the occurrence of STEMI-related arrhythmic and hemodynamic complications in patients treated by pPCI. The data was collected from the consecutive patients treated by pPCI for STEMI at the Emergency Institute for Cardiovascular Diseases and Transplantation. The results  suggest that a combination of 34 acute and chronic risk factors could provide additional benefit in identifying patients at risk of 35 STEMI-related complications, who could benefit from closer follow-up and more intensive prophy- 36 lactic and therapeutic strategies.

Overall, the information presented in this article is informative and I approve its publication after some minor updates. 

Minor comments: I suggest that these comments to be updated before publication.

1.       Increase the font size of Figure 1 graph legends.

2.       English language is good, but minor check is required for sentence construction.

Round 2

Reviewer 1 Report

The updated manuscript is much improved on the previous version. I am
pleased that the authors have implemented all given recommendations. Therefore I recommend to accept the manuscript in the present form.